# A Microbial Co-Culturing System for Producing Cellulose-Hyaluronic Acid Composites

**DOI:** 10.3390/microorganisms11061504

**Published:** 2023-06-05

**Authors:** Marcello Brugnoli, Ilaria Mazzini, Salvatore La China, Luciana De Vero, Maria Gullo

**Affiliations:** 1Unimore Microbial Culture Collection Laboratory, Department of Life Sciences, University of Modena and Reggio Emilia, 42124 Reggio nell’Emilia, Italy; marcello.brugnoli@unimore.it (M.B.); ilaria.mazzini@unimore.it (I.M.); salvatore.lachina@unimore.it (S.L.C.); luciana.devero@unimore.it (L.D.V.); 2NBFC—National Biodiversity Future Center, 90133 Palermo, Italy

**Keywords:** bacterial cellulose, hyaluronic acid, co-culture, acetic acid bacteria, lactic acid bacteria, sustainable composites

## Abstract

In this study, a co-culture system combining bacterial cellulose (BC) producers and hyaluronic acid (HA) producers was developed for four different combinations. AAB of the genus *Komagataeibacter* sp. and LAB of the *Lactocaseibacillus* genus were used to produce BC and HA, respectively. Fourier-transform infrared spectroscopy, scanning electron microscopy, and X-ray diffraction were used to investigate changes in BC-HA composites chemical and morphological structure. Water absorption, uptake, and antibacterial properties were also tested. Outcomes highlighted a higher bacterial cellulose yield and the incorporation of hyaluronic acid into the composite. The presence of hyaluronic acid increased fiber dimension—nearly doubled for some combinations—which led to a decreased crystallinity of the composites. Different results were observed based on the BC producer and HA producer combination. However, water holding capacity (WHC) in all the samples improved with the presence of HA, while water uptake worsened. A thymol-enriched BC-HA composite showed high antibacterial activity against *Escherichia coli* DSM 30083^T^ and *Staphylococcus aureus* DSM 20231^T^. Results could contribute to opening new applications in the cosmetics or pharmaceutical fields.

## 1. Introduction

Microbial polymers have gained tremendous attention over the decades owing to their outstanding properties and versatility. Among microbial polymers, bacterial cellulose (BC) stands out as one of the most versatile biopolymers. It is produced by many bacteria, of which acetic acid bacteria (AAB) of the genus *Komagataeibacter* are the most efficient.

However, even though BC possesses excellent properties, it lacks several features such as an antibacterial effect, an antiaging effect, elasticity, or good optical properties, which limits its potential uses [1]. For this reason, several studies have been focused on improving BC properties or adding new functional features by adding molecules through different functionalization strategies [2,3,4,5,6]. BC functionalization could be performed through ex-situ or in-situ modifications and multi-microbial systems [7,8].

Ex-situ modifications consist of adding the chosen substance to BC after its formation, by immersing the BC pellicle into a solution containing the additional molecule. The porosity of BC allows the selected molecule to spread through the matrix of fibers and facilitate absorption [9]. On the contrary, during in-situ modification, different additives are provided by manually adding them to the culture medium while BC is forming, or at the time of the starting inoculum [7,10]. For example, BC has been supplemented with plant polymers [11], synthetic polymers [12], proteins [13,14], or water-soluble polymers like alginate [15]. Any added material can alter the shape, structure, and properties of BC. A composite formed by BC and egg-white showed increased store modulus [13], a BC-lignin blend exhibited improved stiffness and water resistance performance [11], and BC-alginate membranes showed high water absorption and a significant decrease in O_2_ permeability [15]. However, many polymers are naturally secreted by microorganisms, which could be integrated into a BC matrix [8], limiting extraction and purification steps. For this reason, instead of using purified materials, in recent years, multi-microbial systems, namely co-culture, have emerged, as cost-effective strategies. In a microbial co-culture system, microorganisms are cultivated together, obtaining combined composites, directly during microbial growth [16].

Hyaluronic acid (HA) is widely distributed in nature, both within eucaryotic and prokaryotic organisms. However, microbial HA, synthetized by wild type and engineered bacteria, is of industrial relevance. Among LAB, several species are reported as HA producers [17,18,19,20]. HA is well-known for its tissue repair and angiogenesis properties [21]. BC-HA scaffolds have already been investigated for potential applications in the biomedical field, such as wound dressing and tissue regeneration scaffolds [22,23,24]. Tang and co-workers [24] incorporated HA into BC by immersing the latter in a solution of HA and a cross-linking agent. On the contrary, through an in-situ modification, Lopes and co-workers [22] added hyaluronic acid to a culture of acetic acid bacteria (AAB) at the beginning of the bacterial growth, achieving a BC-HA composite with additional features compared to pure BC.

In this study, a co-culture system was implemented as an alternative to the previously mentioned strategies for developing a BC-HA composite with potential applications in the biomedical or cosmetic fields. BC-HA composites produced by four combinations of AAB-LAB pairs showed different concentrations of HA. In addition, an increase in BC yield was observed in co-culture conditions. Scanning electron microscopy (SEM), X-ray diffractometry (XRD), and Fourier transform infrared spectroscopy (FT-IR) were assessed to evaluate changes compared to pure BC, whereas water absorption and release tests were performed to evaluate BC-HAs water properties. Finally, thymol-enriched BC-HA antibacterial activity was tested against *E. coli* and *S. aureus*.

## 2. Materials and Methods

### 2.1. Bacterial Strains

In this study, bacterial strains from UMCC (Unimore Microbial Culture Collection, Reggio nell’Emilia, Italy) and DSMZ (Deutsche Sammlung von Mikroorganismen und Zellkulturen, Braunschweig, Germany) were used. *K. xylinus* UMCC 2947 and *Komagataeibacter* sp. UMCC 3071 were chosen among AAB, *L. rhamnosus* UMCC 2496 and *L. casei* UMCC 2535 were chosen among LAB, and *E. coli* (DSM 30083^T^) and *S. aureus* (DSM 20231^T^) were used to test antibacterial activity.

### 2.2. Co-Culture Procedure

The AAB and LAB strains used in this study were rehydrated from −80 °C storage conditions. AAB were cultivated in Hestrin-Schramm medium [25] (20.00 g/L glucose anhydrous, 10.00 g/L yeast extract; 5.00 g/L polypeptone; 2.70 g/L disodium phosphate anhydrous; and 1.15 g/L citric acid monohydrate) (HS) at 28 °C for 4 days. LAB were cultivated in De Man, Rogosa, and Sharpe medium (MRS) [26] (20.00 g/L glucose, 10.00 g/L peptone, 10.00 g/L beef extract, 5.00 g/L yeast extract, 2.00 g/L dipotassium hydrogen phosphate, 5.00 g/L sodium acetate, 2.00 g/L ammonium citrate, 0.20 g/L magnesium sulfate, 0.05 g/L manganous sulfate, Tween© 80) and L-cysteine (Sigma-Aldrich, Milan, Italy) (0.50 g/L) at 30 °C for 2 days. The co-culture procedure was performed in flasks using HS-modified medium (30.00 g/L glucose anhydrous, 10.00 g/L yeast extract; 5.00 g/L polypeptone; 2.70 g/L disodium phosphate anhydrous, 0.50 g/L L-cysteine, 0.20 g/L magnesium sulfate, 0.05 g/L manganese sulfate). The AAB inoculum was set at a standard volume (5% *v*/*v*), whereas the LAB inoculum was set at a cell’s concentration of 10^6^ CFU/mL. The LAB cell’s concentration was standardized by measuring the optical density of the cultures at 600 nm (OD_600_) using a UV-6300PC spectrophotometer (VWR, Dallas, TX, USA). Flasks were incubated at 30 °C for 72 h in static conditions. Tests were conducted in triplicate.

### 2.3. Harvesting, Purification, and Quantification of Bacterial Cellulose

BC layers were removed from the culture broth, washed two times in distilled water, and incubated in a 0.1 M NaOH solution at 80 °C for 30 min. BC was placed in shaking conditions overnight to remove cells and residual NaOH, and then dried in an oven at 20 °C until it reached a constant weight. The weighting of the dried BC layer was performed using an analytical balance (Gibertini E42S, sensitivity 0.1 mg, Gibertini, Milan, Italy).

The final BC yield value expressed as grams of dried BC per liter of medium (g/L) is the average of three biological replicates.

### 2.4. Extraction and Quantification of Hyaluronic Acid from LAB Cultures

The extraction of HA was performed according to the protocol used by Liu and Catchmark [27] and Mohan and co-workers [28], with some modifications.

Briefly, 1% *v*/*v* of a 5% (*w*/*v*) sodium dodecyl sulfate (Sigma-Aldrich, Milan, Italy) solution was added to liquid cultures. Tubes were incubated for 10 min at room temperature and centrifuged (8000× *g* at 4 °C for 5 min). The supernatant was mixed with an equal volume of 1.7% (*w*/*v*) of cetylpyridinium chloride (CPC) (Sigma-Aldrich, Milan, Italy). The CPC-HA precipitate was harvested by centrifugation (8000× *g*/10 min) and resuspended in 2 mL 1 M NaCl solution. To release HA from CPC, the solution was incubated in a water bath at 55 °C for 10 min. Undissolved material was removed by centrifuging (8000× *g*/10 min). Then, three volumes of ethanol were added to one volume of the resuspended solution and centrifuged (8000× *g*/10 min). The gathered HA was resuspended in distilled water and stored at 4 °C, until use.

HA quantification was performed using the cetyltrimethylammonium bromide turbidimetric method (CTM) following Song and coworkers procedure [29]. Briefly, CTM was prepared by dissolving 2.5 g of cetyltrimethyl ammonium bromide (CTAB) in a solution of NaOH 2% *w*/*v* and heating to 37 °C. All96-well microplates were filled with 50 μL of acetate buffer (pH 6) and 50 μL of the dissolved pellet of the sample. For the blank, wells were filled with 100 μL of acetate buffer. Microplates were incubated at 37 °C for 15 min. After that, 100 μL of CTM reagent was added to each well. Microplates were then shaken for 10 s and incubated at 37 °C for 10 min. Absorbance was read at 400 nm (UV-6300PC, VWR, Dallas, TX, USA). The HA concentration (mg/mL) was calculated using the calibration curve. Each determination was performed in triplicate.

### 2.5. Hyaluronic Acid Content of BC-HA Composite

Dried BC-HA composites (area: 90 cm^2^) were immersed in water with 1% *w*/*v* of cellulase from *Trichoderma viride* (Sigma-Aldrich), and pH was adjusted at 5, following Takahama and co-workers [23] procedure. Sample digestion was performed at 37 °C for 72 h. The supernatants were separated from the residues through centrifugation (10,000× *g*/10 min at 4 °C). The HA in each supernatant was purified by ethanol precipitation and quantified with the CTM method, as previously described. The amount of HA was reported as milligrams of HA per gram of BC-HA composite (mg/g).

### 2.6. Characterization of BC-HA Composites

#### 2.6.1. Fourier Transform Infrared Spectroscopy

FT-IR spectra were acquired using a Vertex 70 (Bruker, Bremen, Germany) spectrometer equipped with an attenuated total reflection (ATR) Golden Gate diamond sensor. The wavenumber range was set from 4000 to 400 cm^−1^ with an accumulation of 32 scans and a resolution of 4 cm^−1^.

#### 2.6.2. Scanning Electron Microscopy

To perform SEM analysis, samples were cut (1 × 1 cm^2^), coated with a thin layer of gold (Au), and mounted on a stainless-steel stub with double-sided tape. A field emission Nova NanoSEM 450 (Bruker, Germany) operating at 10 kV in high vacuum conditions was used to carry out SEM analysis.

#### 2.6.3. X-ray Diffraction Pattern and Crystallinity

XRD patterns were obtained using a Panalytical X’Pert PRO (Malvern Panalytical, Malvern, Worcestershire, UK) diffractometer with CuKα radiation source (λ = 1.54 Å) at a voltage of 40 kV and a filament emission of 40 mA.

Samples were placed on a zero-background sample holder to avoid the detection of any peak not related to the samples. Samples’ diffracted radiation intensity was measured between 10 and 30° (2θ) with ramping at 1°/min. The crystallinity index (CI) was calculated using the following formula:CI (%) = (sc/st) × 100 (1)
where sc is the sum of the diffraction peak area and st is the sum of the total area.

### 2.7. Water-Uptake Assay

A water-uptake assay was performed on dried samples (diameter: 7.5 cm), following Morais and co-workers [4] procedure. Briefly, dried samples were weighted and soaked in distilled water. Each sample was placed in an individual container. After three hours, excessive water was gently removed with absorbent paper, and then samples were weighted. Water uptake was calculated as follows:Water uptake (%) = ((Ww − Wd)/Wd) × 100 (2)
where Wd and Ww are the weights of dried and wet samples, respectively.

The values were expressed as the average ± standard deviation of three biological replicates.

### 2.8. Water Holding Capacity Assay

The samples WHC was determined by using wet disks of 7.5 cm in diameter. The weights of the samples were recorded periodically after excessive water was gently removed with absorbent paper. Weighting was performed until a constant weight was reached. The WHC was calculated as follows:WHC (%) = ((Wrw − Wd)/(Ww − Wd)) × 100 (3)
where Wrw is the weight of the sample, Wd is the weight of the dried sample, and Ww is the weight of the sample at the beginning of the test.

The values were expressed as the average ± standard deviation of three biological replicates.

### 2.9. Preparation of Thymol-Enriched Composite and Antibacterial Activity Test

To evaluate the antibacterial activity, BC-HA disks (diameter: about 1.5 cm) were immersed in an aqueous solution containing 10 mg/mL of thymol (Extrasynthese, Genay, France). BC-HA disks were soaked with sterile water and used as negative controls, while sterile paper disks loaded with thymol solution were used as positive controls. The antibacterial activity of thymol-enriched BC-HA was analyzed using the disk-diffusion method. The inhibition activity was tested against *Escherichia coli* DSM 30083^T^ and *Staphylococcus aureus* DSM 20231^T^. The strains were cultivated at 30 °C in Brain Heart Infusion Broth (27.5 g/L of brain heart infusion and peptones, 2 g/L of glucose, 5 g/L of sodium chloride, and 2.5 g/L of disodium hydrogen phosphate). Then, the antibacterial activity test was performed by spreading bacterial suspensions containing10^6^ CFU/mL cells on the surface of plate count agar media (PCA) (5 g/L of tryptone, 2.5 g/L of yeast extract, 1 g/L of glucose, and 15 g/L of agar). Petri dishes were incubated upside-down for 24 h at 37 °C. Finally, the microbial inhibition zone was measured through the utilization of imageJ 1.53k software [30].

### 2.10. Data Statistical Analysis

Experimental data were analyzed using R v 4.2.3 [31] at a significance level of *p* = 0.05 and reported as the average of the triplicate ± standard deviation. The statistical significance was determined using one-way ANOVA, and the Tukey post-hoc test was used to determine statistical differences among samples.

## 3. Results

### 3.1. Bacterial Cellulose and Hyaluronic Acid Production by AAB and LAB in Monoculture Conditions

The selection of strains able to produce BC was done exclusively with strains belonging to the *Komagataeibacter* genus. This was due to the evidence that *K. xylinus* and closely related species are reported as high BC producers [32,33,34]. UMCC 3071 resulted in the highest producer, exceeding 2 g/L of BC (Table 1). Instead, HA producers were chosen within *L. casei*, and *L. rhamnosus* species according to a literature review [18,19,35,36,37]. Among *L. casei* UMCC 2535 and *L. rhamnosus* UMCC 2496, the latter presented the highest yield of HA, as reported in Table 1.

### 3.2. Bacterial Cellulose and Hyaluronic Acid Production in Co-Culture System

AAB and LAB strains were co-cultured through 4 different combinations of AAB-LAB pairs (Table 2), namely UMCC 2947-UMCC 2535 (C1), UMCC 2947-UMCC 2496 (C2), UMCC 3071-UMCC 2535 (C3) and UMCC 3071-UMCC 2496 (C4).

All the combinations showed the presence of a cellulosic layer on the surface. The BC yield reached in each combination is shown in Figure 1.

Compared to the monoculture system, higher production of BC was achieved in the co-culture system, independently from the AAB-LAB combination. Indeed, with respect to production in the monoculture system, BC production increased by 86% in C4 composite and 64% in C1, compared to pure BC produced by UMCC 3071 and UMCC 2947, respectively. The highest BC yield was observed in C1, reaching 3.44 g/L.

HA content (Figure 2) in C1 and C2 composites resulted in 2.00 and 2.10 mg HA/g dried BC, respectively. The HA content in the C4 composite was moderately higher, resulting in 3.42 mg/g dried BC. The highest HA content was detected in the C3 combination, reaching almost 10 mg/g dried BC.

### 3.3. Characterization of Bacterial Cellulose-Hyaluronic Acid Composite

The chemical structure of BC samples and the presence of HA in BC-HA composites were investigated by ATR-FTIR measurements (Figure 3).

Both pure BC obtained in monoculture systems and BC-HA samples’ spectra presented characteristic absorption bands of BC functional groups at around 3340, 2895, 1423, 1315, 1154, and 1040 cm^−1^ [5,40,41,42].

The spectrum of HA showed a wide band at about 3412 cm^−1^ and bands of moderate intensity at around 2916 cm^−1^ [22]. Absorption bands at around 1604, and 1400 cm^−1^ indicate carboxylate asymmetric stretching vibration and carboxylate symmetric stretching, respectively [43]. Signals at 1560, and 1322 cm^−1^ are assigned to amide II, and amide III, respectively [44].

All BC-HA composite spectra, excluding C1, showed an intense peak of secondary amide N-H bending and C-N stretching at around 1550 cm^−1^ [45]. On the other hand, only in C1, we observed a peak at 1728 cm^−1^, which was absent even in HA or pure BC spectra. Finally, for all BC-HA samples, a strong vibrational band at around 1640 cm^−1^ (N-H bending amide II and COO- asymmetric stretching) is clearly visible.

The diffraction diagrams of pure BC and BC-HA composites (Figure 4) correspond to the profile of cellulose I, with reflections at 15°, 17°, and 23.2° indexed as 100, 010, and 110 crystallographic planes. The XRD profiles of BC and BC-HA were similar, except for the C1 composite, which showed peaks with higher intensity.

The CI slightly varied for combinations (C1 and C2) involving UMCC 3071 as BC producer (Table 3), ranging between 73 and 76%. On the other hand, the crystallinity of BC produced by UMCC 2947 was mainly affected when produced in co-culture, decreasing from 88% to 84% and 80% when produced by UMCC 2535 and UMCC 2496, respectively. Generally, co-culture systems and HA presence lead to a decrease in CI in all BC-HA samples.

SEM images revealed that all BC layers and BC-HA composite surfaces were characterized by a net of BC microfibrils casually assembled (Figure 5).

When AAB were cultivated with the LAB strain UMCC 2535 (C1, C3), BC-HA presented a network of microfibrils intertwined with large ribbons (Figure 5c,d), whereas no variation in fiber arrangement was observed in the presence of UMCC 2496. A comparison of the results showed considerable increases in the ranges of microfibril diameters between BC-HA and pure BC (Appendix A). C3 and C4 showed the highest fiber diameters, reaching 110.57 nm and 110.29 nm, respectively. Interestingly, the C3 sample (Figure 5d) was the only one presenting a bimodal distribution of fiber diameters. Indeed, almost all of the fibers presented a diameter higher than 130 nm or lower than 100 nm, with a few fibers ranging between 100 and 130 nm.

### 3.4. Bacterial Cellulose-Hyaluronic Acid Composite Water Absorption and Release Properties

The rehydration ability of the BC-HA composites and pure BC was evaluated by immersing the samples in water for 3 h at room temperature. The water uptake percentage is reported in Figure 6.

Generally, the co-culture system and HA presence worsened the water uptake capacity of composites compared to pure BC. However, a considerable difference could be observed among composites produced by different AAB strains. Indeed, UMCC 3071 BC-HA composites absorbed more water than any combination where UMCC 2947 was involved.

In accordance with Tang and co-workers [24], the presence of HA inside BC increased the WHC of all the samples. C1 resulted in the composite with the lower release rate of water, retaining 75% of it after 24 h. Besides HA effect, no variations related to LAB species seemed to have occurred. Interestingly, differences in WHC patterns of pure BC produced by UMCC 2947 and UMCC 3071 were observed. UMCC 3071 BC lost water more rapidly, retaining after 6 h 79% of water whereas UMCC 2947 almost 86%. Differences could be related to BC fibrous network structure. Indeed, BC with higher fiber density tends to retain a higher percentage of liquid. On the contrary BC with low fiber density presents more empty space among the fibers, which leads a higher absorption of liquid but a lower holding capacity.

### 3.5. Antibacterial Activity of Thymol-Enriched Bacterial Cellulose-Hyaluronic Acid Composite

To test the antibacterial properties, a disk diffusion test was conducted on the C1 sample by immersing it in a 1% *w*/*v* thymol solution. Growth inhibition was noted with both thymol-enriched paper disks and thymol-enriched BC-HA composites (Table 4). No activity of BC-HA control was observed against either *E. coli* or *S. aureus*. BC-HA exhibited a wider inhibition zone compared to the positive control, reaching an inhibition zone of 22.17 mm and 30.12 mm against *E. coli* and *S. aureus*, respectively. Therefore, the incorporation of thymol into BC-HA showed great antibacterial activity against *S. aureus*, which is one of the most common isolates present in burn wounds [46]. Results are in accordance with Jiji and co-workers [47], who tested a thymol-enriched BC layer against various pathogens.

## 4. Discussion

In this study, a microbial co-culture system using AAB and LAB strains was developed to produce BC-HA composites.

AAB strains UMCC 2947 and UMCC 3071 were previously studied in the context of works aimed at recovering and selecting AAB able to produce BC [38,48,49,50]. Results obtained in the present study are in agreement with our previous studies reporting BC production as a strain-specific trait within the *Komagataeibacter* genus [50,51].

Among LAB strains, even though they had a lower yield compared to other studies [35,36], both UMCC 2535 and UMCC 2496 produced valuable amounts of HA. However, LABs capability to produce HA has been reported as a strain-dependent trait [36]. Probably the different behavior is related to the variability in the transport efficiency across the membrane and on the glucose uptake system [18].

From all the AAB-LAB co-culture combinations (C1, C2, C3, and C4), production of BC-HA was obtained. The highest BC yield was observed in the C1 sample, reaching 3.44 g/L. It’s worth noting that AAB strains produced more BC when co-cultivated with both LAB strains, compared to BC obtained by the monoculture (Figure 2). Similar results were previously reported in other studies, where different co-culture systems provided higher BC yields relative to the monoculture. [52,53,54]. Seto and co-workers [52] reported enhanced BC production by *G. xylinus* st-60-12 when co-cultured with *L. mali* st-20. This evidence has been attributed to a facilitated coaggregation of cells, supporting the assembly of fibers into BC by LAB [52]. In addition, Jiang and co-workers studied the effect of co-culturing *K. nataicola* with 17 strains of *Lactobacillus* spp. to produce BC. Compared with *K. nataicola* monoculture, 5 combinations with LAB showed an increase of BC yield (from 23.1% to 59.5%) [54]. The authors suggested that the production of BC in a co-culture system is enhanced by the more efficient formation of the intracellular β-1,4-glucan bond, and by the presence of lactic and acetic acids, which turn on the Krebs cycle.

Furthermore, it has been observed an increased BC yield (10.8%) and improved mechanical properties by a *K. hansenii* strain co-cultured with *E. coli* [53].

In our study, composites involving the AAB UMCC 3071 had the highest HA content (C3 and C4). On the other hand, no significant differences were observed in C1 and C2 (Figure 2).

The chemical structure, morphology, and crystal structure of BC-HA composites were analyzed and compared with those of pure BC obtained by the same strains in monoculture systems. The ATR-FTIR spectra (Figure 3) presented characteristic absorption bands of BC functional groups in all the samples. BC-HA samples’ spectra presented peaks that confirmed HA presence such as the intense peak at around 1550 cm^−1^, attributable to secondary amide N-H bending and C-N stretching [45]. In addition, all BC-HA composites and pure BC spectra presented a band at around 1640 cm^−1^ associable with minor water residues. However, the higher intensity in BC-HA samples could be due to N-H bending amide II and COO- asymmetric stretching, as already reported [27,41]. Contrary to the C2, C3, and C4 composites, sample C1 was the only one presenting a peak at 1728 cm^−1^. This peak is attributable to the protonated form of HA in BC-HA samples due to linkage between the hydroxyl group of BC and the carboxyl group of HA, as previously observed [22]. Indeed, no peaks at around 1728 cm^−1^ were detected in the HA sodium salt used as a standard.

Changes in the chemical structure and morphology of composites lead to a reduction of CI in all the samples (Table 3), with C1 and C2 composites being the most affected. The decrease in crystallinity could be attributed to the amorphous form of HA present in the BC network, as reported by Lopez and co-workers [43]. However, the nearly unchanged XRD patterns and the light reduction of crystallinity indicated that the presence of HA slightly interfered with the self-assembly of the fibrous network, as confirmed by SEM analysis. Indeed, a net of fibers casually assembled for pure BC and BC-HA composites was observed (Figure 5), according to previous studies [14,24,55]. An increase in average fiber diameter was observed in BC-HA samples compared to pure BC. Similar results (ribbon width of BC fibers ranging between 103 and 125 nm) were previously obtained by co-cultivating an engineered *Lactococcus lactis* strain with a *K. hansenii* strain [56]. Lopes and co-workers [22] also observed an increase in ribbon width to over 100 nm in BC-HA hybrid membranes obtained by adding HA into the culture medium during the BC synthesis. According to Tang and co-workers [24], HA tends to associate with BC and bundle BC fibers, resulting in larger fibers, as we observed in the C3 composite. In addition, as hypothesized by Chi and Catchmark [57], the polysaccharide location on the surface of microfibrils affects the fiber diameter. The authors observed BC bundles with double width when BC was obtained by adding polysaccharides to the culture medium.

Water absorption and WHC are two important parameters considering the biomedical and cosmetic uses of biopolymers [24,58]. The swelling of BC-HA facilitates the delivery of HA in potential sheet facial masks or antiaging patches. Moisture retention is a highly desirable property for any on-skin application (i.e. wound dressing) aiming to reduce the dehydration of the patch and ensure its adherence to the skin [59]. Water uptake and holding capacity are strictly dependent on BC fiber arrangement. Pellicles with well-developed voids tend to release water rapidly, while denser fiber structures will slow the release [60]. On the contrary, water uptake will be higher with well-developed void structures. In this study, the highest absorption of water was reached in C4 and C3 composites. However, generally, when the AAB UMCC 3071 was used in co-culture, composites had a higher water uptake and a lower crystallinity. Indeed, an increase in amorphous state is associated with a higher absorption capacity [61]. In addition, a high fiber density or amount of BC could prevent the absorption of liquids [43]. In our study, C1 and C2 composites exhibited lower absorption of HA compared to C3 and C4.

WHC increased in all BC-HA samples independently of the LAB strain involved in the co-culture. As observed by Tang and co-workers [24], the reason could be due to the presence of HA, which displays high WHC.

Wound dressing materials should maintain moisture for a prolonged period, slowly release any loaded active molecules, and have effective antibacterial activity against the main pathogenic microorganisms harmful to the wound [47,62]. For these reasons, we performed a preliminary antibacterial test against representative pathogenic gram+ and gram- bacteria. The test was conducted on C1 composite, which resulted in the sample with the highest WHC, by immersing the sample in a thymol solution.

Thymol-enriched C1 composites showed great antibacterial properties against both *E. coli* and *S. aureus*, significantly higher than thymol paper disks (Table 4). The enhanced antibacterial activity may be due to the well-dispersion of thymol in the BC-HA structure. Indeed, the fiber arrangement, porosity, and density strictly influence the leaching and release properties of BC layers [63].

*S. aureus* DSM 20231^T^ was more affected by the antibacterial activity of the BC-HA composite compared to *E. coli* DSM 30083^T^. This could be due to gram-negative and gram-positive differences in the outer cell walls and membranes. Indeed, gram negative bacteria possess detoxifying enzymes in the periplasmic space and a hydrophilic outer membrane, which could exert protective effects, whereas gram-positive bacteria do not possess neither outer membrane nor protective enzymes in the periplasmic space [64].

## 5. Conclusions

In this study, a co-culture system by AAB and LAB allowed the production of BC-HA composites with essential features for potential biomedical and cosmetic applications.

Based on our results, both AAB and LAB strains can be effectively co-cultivated under the right media and conditions. Produced BC-HA composites have outstanding potential in the cosmetic field as beauty masks or patches for face anti-aging and hydrating treatments. Both biopolymers contribute to the main composite properties. BC provides adhesiveness and moisture to the skin, while HA could act as a filling and hydrating agent. Moreover, preliminary tests showed that thymol enriched BC-HA possesses antibacterial activity against gram-positive and gram-negative bacteria. This evidence highlights the possibility of using such composites as wound dressing materials for burn wound repair. Further tests will be performed to evaluate the cytotoxicity of the composites and the release kinetics of HA.

Finally, in this study, the suitability of a microbial co-culturing system was demonstrated, opening new perspectives for developing next-generation composites based on green-friendly cellulose combined with hyaluronic acid.

## Figures and Tables

**Figure 1 microorganisms-11-01504-f001:**
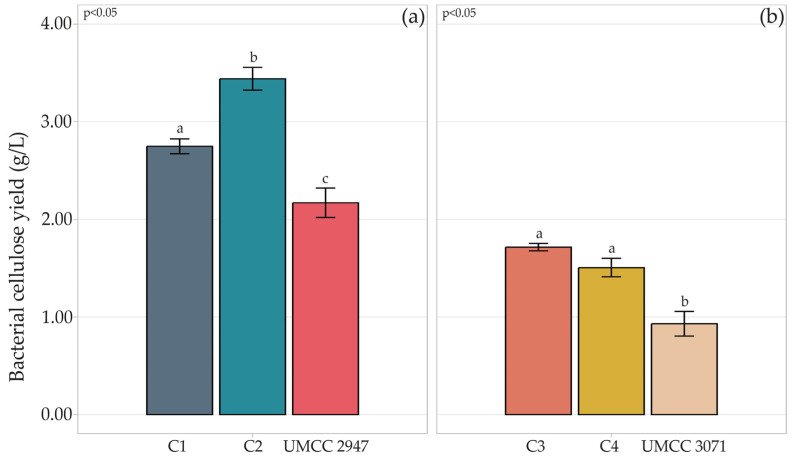
Quantification of BC (g/L) produced by strains UMCC 2947 (**a**) and UMCC 3071 (**b**) in monoculture and in combination with LAB. Bar plots indicate the average BC by three replicates ± standard deviation. Significant differences among BC yields are shown by different letters (*p* ≤ 0.05).

**Figure 2 microorganisms-11-01504-f002:**
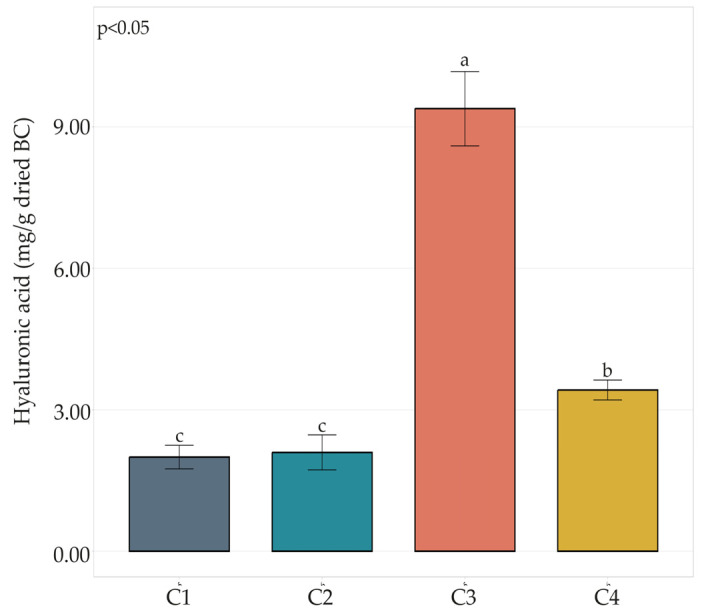
HA content in BC-HA composites C1, C2, C3, and C4. Bar plots indicate the average HA content by three replicates ± standard deviation. Significant differences among HA yields are shown by different letters (*p* ≤ 0.05).

**Figure 3 microorganisms-11-01504-f003:**
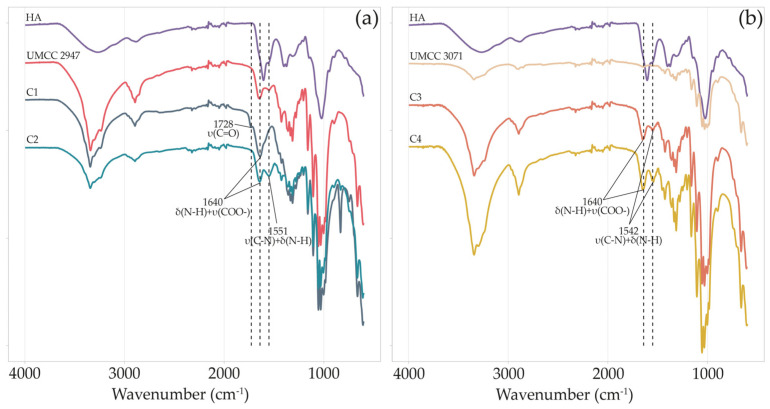
FT-IR spectra of HA, pure BC, and BC-HA composites produced by UMCC 2947 (**a**) and UMCC 3071 (**b**) as monoculture and in combination with LAB. Each color represents a different sample. δ and υ indicate bending and stretching vibrations, respectively.

**Figure 4 microorganisms-11-01504-f004:**
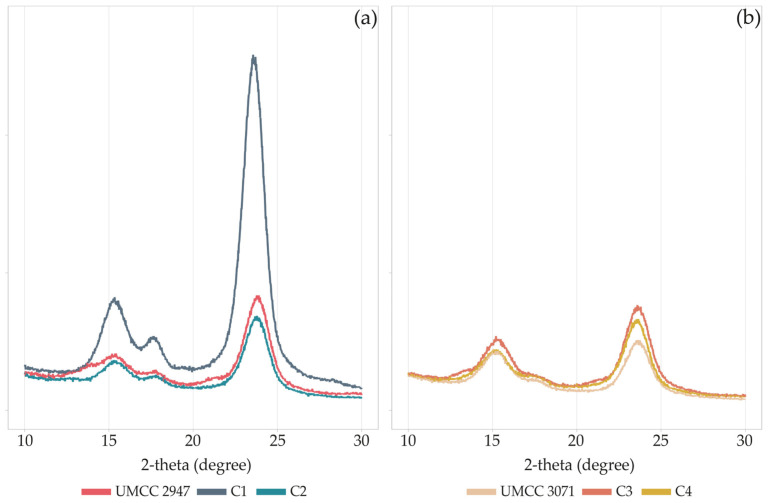
XRD patterns of pure BC and BC-HA composites produced by UMCC 2947 (**a**) and UMCC 3071 (**b**) as monocultures and in combination with LAB. Each color represents a different sample.

**Figure 5 microorganisms-11-01504-f005:**
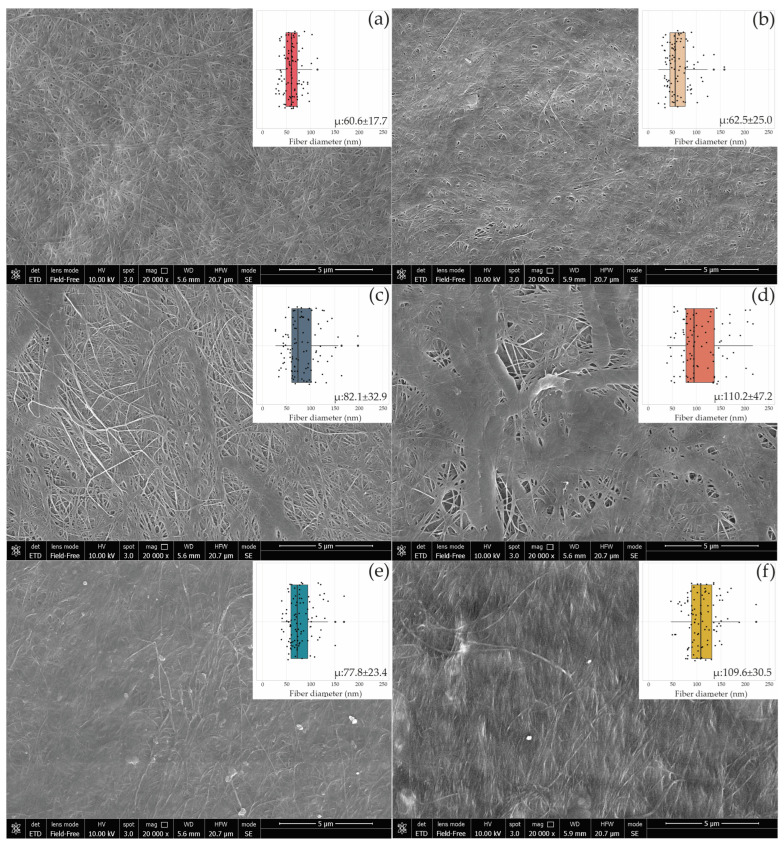
SEM images of the BC surface produced by UMCC 2947 (**a**), UMCC 3071 (**b**), and BC-HA composites C1 (**c**), C2 (**e**), C3 (**d**), and C4 (**f**). The boxplot in the inserts shows the fiber diameter distribution in each sample synthesized in monoculture (**a**,**b**) and co-culture with UMCC 2535 (**c**,**d**) and UMCC 2496 (**e**,**f**). “μ” indicates the average diameter (nm) ± standard deviation of 100 randomly chosen fibers of bacterial cellulose microfibrils.

**Figure 6 microorganisms-11-01504-f006:**
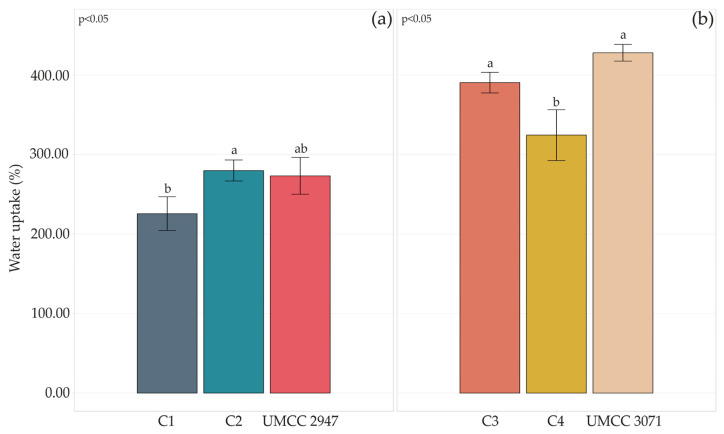
Water uptake percentage after 3 h of pure BC and BC-HA composites produced by UMCC 2947 (**a**) and UMCC 3071 (**b**) as monocultures and in combination with LAB. Each color represents a different sample. Bar plots indicate the average water uptake by three replicates ± standard deviation. A significant difference among water uptake is shown by different letters (*p* ≤ 0.05).

**Table 1 microorganisms-11-01504-t001:** BC production of UMCC 2497 and UMCC 3071 and HA production of UMCC 2496 and UMCC 2535.

**Strain Designation**	*** Species**	**BC (g/L)**
UMCC 2947	*K. xylinus*	1.99 ± 0.01
UMCC 3071	*Komagataeibacter* sp.	2.34 ± 0.02
**Strain designation**	**** Species**	**HA (mg/mL)**
UMCC 2496	*L. rhamnosus*	0.216 ± 0.024
UMCC 2535	*L. casei*	0.198 ± 0.022

* [38]; ** [39].

**Table 2 microorganisms-11-01504-t002:** Codification of AAB-LAB co-culture systems.

AAB	LAB	
	**UMCC 2535**	**UMCC 2496**
**UMCC 2947**	C1	C2
**UMCC 3071**	C3	C4

**Table 3 microorganisms-11-01504-t003:** Crystallinity Index of BC and BC-HA composites.

Samples	Crystallinity Index
UMCC 2947	88%
C1	84%
C2	80%
UMCC 3071	76%
C3	73%
C4	74%

**Table 4 microorganisms-11-01504-t004:** Antibacterial activity of C1 composite, thymol enriched C1, and thymol enriched paper disk.

Bacterial Species	Zone of Inhibition (mm)		
	C1	Thymol C1	Thymol Paper Disk
*E. coli* DSM 30083^T^	0	22.17 ^a^ ± 1.20	16.26 ^b^ ± 0.30
*S. aureus* DSM 20231^T^	0	30.12 ^a^ ± 1.69	15.74 ^b^ ± 0.45

Data are expressed as means ± standard deviations. Different letters indicate statistical differences within the same row at *p* ≤ 0.05.

## Data Availability

All data underlying the results are included as part of the published article.

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
