# Peer review of "A Microbial Co-Culturing System for Producing Cellulose-Hyaluronic Acid Composites"

_microorganisms, 2023, doi:10.3390/microorganisms11061504_

Round 1

Reviewer 1 Report

The co-culture of strains for BC and HA composite is an interesting work. The BC or HA production in composite is higher than that in monoculture. It is acceptable after minor revision.

1. line 244, C4 may be C3

Reviewer 2 Report

Brugnoli et al. presented a co-culture system for efficiently producing bacterial cellulose (BC) and hyaluronic acid (HA) composite and also a method for preparing  thymol-enriched BC-HA composite was derived for high antibacterial applications. The experiments were well-designed and the properties of the composites were thoroughly characterized. There are still some suggestions for this manuscript before it’s recommended for publishing,

1.    The manuscript was composed of dedicated Results and Discussion sections, it is more convenient for the readers to understand the research outcome if the discussion of each characterization is placed right after its corresponding Results paragraph. Also, a Conclusion section for summarizing the research findings in the work is necessary to be included as the last section.

2.    In table 4, the thymol paper disk showed less antibacterial property compared to thymol C1. Considering the pristine C1 sample was not antibacterial, what is the major cause of the increase in antibacterial activity? Is there any synergistic effect or the C1 composite membrane can boost the thymol release kinetics? Please specify.

3.     In page 11, line 333, please change 3.01 cm to 30.1 mm so as to be consistent with the other unit.

Reviewer 3 Report

The submitted work tittled “A microbial co-culturing system for producing cellulose-hyaluronic acid composites” by Brunoli, M.; et al. is an original scientific work where the authors fully characterize blends made of cellulose and hialuronic acid. Moreover, four different combinations were tested in order to unravel what renders better performance in terms of chemistry, water uptake and antimicrobial properties.

However, it exists some points that need to be addressed (please, see them below detailed point-by-point). Here, there exists some suggestions in order to improve the scientific quality of the manuscript paper:

1) KEYWORDS (OPTIONAL). The authors should consider to add the term “sustainable composites” and “antibacterial properties” in the keyword list.

2) INTRODUCTION. “Among microbial polymers, BC stands out (…)” The authors should define the term “bacterial cellulose” in this statement even if it was previously reported in the respective abstract section. Then, the abbreviation should be placed between brackets. Same comment for “HA” (line 50).

3) “However, even though BC possesses excellent properties (…) limits potential uses” (lines 34-36). Here, it may be convenient to add a relevant reference according the information provided by the authors [1].

[1] Choi, S.M.; et al. Bacterial Cellulose and Its Applications. Polymers 2022, 14, 1080. https://doi.org/10.3390/polym14061080.

4) “In a microbial (…) obtaining combined composites, (…). HA (…)” (lines 48-50). Here, the authors introduce the interest to develop the next-generation of composites based on green-friendly cellulose combined with hyaluronic acid. Nevertheless, it lacks a brief explanation of the state-of-the-art and how the composites designed in this study can overcome the limitations reported in previous works. In this framework, the authors should cite the possibility to mix cellulose with other plant polymers like lignin [2] or proteins as egg-white [3] to form composites. The first blends exhibit greater stiffness and water resistance performance but poor antioxidant properties, whereas in the second case the store modulus is increased but the formed composite is susceptible to microbial degradation.

[2] Gerbin, E.; et al. Dual Antioxidant Properties and Organic Radical Stabilization in Cellulose Nanocomposite Films Functionalized by In Situ Polymerization of Coniferyl Alcohol. Biomacromolecules 2020, 21, 3163-3175. https://doi.org/10.1021/acs.biomac.0c00583.

[3] Tomcyńska-Mleko, M.; et al. New product development: Cellulose/egg white protein blend fibers. Carbohydr. Polym. 2015, 126, 168-174. https://doi.org/10.1016/j.carbpol.2015.03.008.

5) MATERIALS & METHODS. “(…) (20 g/L glucose anhydrous, 10 g/L yeast extract; 5 g/L polypeptone; 2.7 g/L disodium phosphate anhydrous and 1.15 g/L citric acid monohydrate)” (lines 80-82). Please, the authors should homogenize the significant figures of the provided data. This point should be taken into account in the rest of tha main manuscript body text.

6) RESULTS. Figure 3 (line 250). Even if this information is already explained in the the text (lines 257-266) it may be opportune to insert the most indicative chemical bonds in this Figure. This fact will aid the potential readers to better understand the differences in terms of chemistry of the materials produced by the two types of bacterial strains.

7) Figure 5 (line 286). Please, the authors should show the standard deviation (SD) values related to the average cellulose fibril diameter (µ) for each tested condition. Then, it may be also desirable to carry out a statistical analysis to discern if there exists significant differences between the gathered fibril diameters among the samples (This information could be placed as Supplementary Information, SI).

8) DISCUSSION. The authors perfectly remarks the most relevants outcomes found in this work and how they can benefit for future Industrial applications. No actions are requested for this section.

9) REFERENCES. The references are in the proper format style of Microorganisms. No actions are requested for this section.

The English is fine. Nevertheless, I would recommend the authors to check the English out carefully prior the submission of the revised version

Author Response

Please see the attachment.  In addition, as you suggested, we checked the English out carefully and made some changes to improve it.
